# Weakly Labeled Data Augmentation for Deep Learning: A Study on COVID-19 Detection in Chest X-Rays

**DOI:** 10.3390/diagnostics10060358

**Published:** 2020-05-30

**Authors:** Sivaramakrishnan Rajaraman, Sameer Antani

**Affiliations:** Lister Hill National Center for Biomedical Communications, National Library of Medicine, 8600 Rockville Pike, Bethesda, MD 20894, USA; sameer.antani@nih.gov

**Keywords:** augmentation, chest X-rays, convolutional neural network, COVID-19, deep learning, pneumonia, localization

## Abstract

The novel severe acute respiratory syndrome coronavirus 2 (SARS-CoV-2) has caused a pandemic resulting in over 2.7 million infected individuals and over 190,000 deaths and growing. Assertions in the literature suggest that respiratory disorders due to COVID-19 commonly present with pneumonia-like symptoms which are radiologically confirmed as opacities. Radiology serves as an adjunct to the reverse transcription-polymerase chain reaction test for confirmation and evaluating disease progression. While computed tomography (CT) imaging is more specific than chest X-rays (CXR), its use is limited due to cross-contamination concerns. CXR imaging is commonly used in high-demand situations, placing a significant burden on radiology services. The use of artificial intelligence (AI) has been suggested to alleviate this burden. However, there is a dearth of sufficient training data for developing image-based AI tools. We propose increasing training data for recognizing COVID-19 pneumonia opacities using weakly labeled data augmentation. This follows from a hypothesis that the COVID-19 manifestation would be similar to that caused by other viral pathogens affecting the lungs. We expand the training data distribution for supervised learning through the use of weakly labeled CXR images, automatically pooled from publicly available pneumonia datasets, to classify them into those with bacterial or viral pneumonia opacities. Next, we use these selected images in a stage-wise, strategic approach to train convolutional neural network-based algorithms and compare against those trained with non-augmented data. Weakly labeled data augmentation expands the learned feature space in an attempt to encompass variability in unseen test distributions, enhance inter-class discrimination, and reduce the generalization error. Empirical evaluations demonstrate that simple weakly labeled data augmentation (Acc: 0.5555 and Acc: 0.6536) is better than baseline non-augmented training (Acc: 0.2885 and Acc: 0.5028) in identifying COVID-19 manifestations as viral pneumonia. Interestingly, adding COVID-19 CXRs to simple weakly labeled augmented training data significantly improves the performance (Acc: 0.7095 and Acc: 0.8889), suggesting that COVID-19, though viral in origin, creates a uniquely different presentation in CXRs compared with other viral pneumonia manifestations.

## 1. Introduction

The novel coronavirus disease 2019 (COVID-19) is caused by a strain of coronavirus called severe acute respiratory syndrome coronavirus 2 (SARS-CoV-2) that originated in Wuhan in the Hubei province in China. On 11 March 2020, the World Health Organization (WHO) declared the disease as a pandemic [1], and as of this writing (in late April 2020), there are more than 2.7 million globally confirmed cases with over 190,000 reported deaths with unabated growth. The disease is detected using reverse transcription-polymerase chain reaction (RT-PCR) tests that are shown to exhibit high specificity but variable sensitivity in detecting the presence of the disease [2]. However, these test kits are in limited supply in some geographical regions, particularly third-world countries [3]. The turnaround time is reported to be 24 h in major cities and even greater in rural regions. This necessitates the need to explore other options to identify the disease and facilitate swift referrals for the COVID-19-affected patient population in need of urgent medical care.

A study of the literature shows that individuals suffering from COVID-19 disease commonly present with hyperthermia and difficulty with breathing. The disease manifests in the lungs as ground-glass opacities, with peripheral, bilateral, and predominant basal distribution [2]. These patterns are visually similar to, yet distinct from, those caused by non-COVID-19-related viral pneumonia and those caused by other bacterial and fungal pathogens [2]. Further, the current literature studies revealed that it is difficult to distinguish viral pneumonia from others caused by bacterial and fungal pathogens [4]. Figure 1 shows instances of chest X-rays (CXRs) of clear lungs, bacterial pneumonia, and COVID-19-related pneumonia, respectively.

While not recommended as a primary diagnostic tool due to the risk of increased transmission, chest radiography and computed tomography (CT) scans are used to screen/confirm respiratory damage in COVID-19 disease and evaluate its progression [3]. CT scans are reported to be less specific than RT-PCR but highly sensitive in detecting COVID-19, and can play a pivotal role in disease diagnosis/treatment [3]. However, the American College of Radiology has recommended against the use of CT scans as a first-line test [5]. Additional considerations of the increased risk of transmission, access, and cost also contribute to the recommendation. When radiological imaging is considered necessary, portable chest X-rays (CXRs) are considered a good and viable alternative [2]. However, in a pandemic situation, the assessment of the images places a huge burden on radiological expertise, which is often lacking in regions with limited resources. Automated decision-making tools could be valuable in alleviating some of this burden, and also as a research tool for quantifying disease progression.

A study of the literature shows that automated computer-aided diagnostic (CADx) tools built with data-driven deep learning (DL) algorithms using convolutional neural networks (CNN) have shown promise in detecting, classifying, and quantifying COVID-19-related disease patterns using CXRs and CT scans [2,3,6], and can serve as a triage under resource-constrained settings, thereby facilitating swift referrals that need urgent patient care. These tools combine elements of radiology and computer vision to learn the hierarchical feature representations from medical images to identify typical disease manifestations and localize suspicious regions of interest (ROI).

It is customary to train and test a DL model with the data coming from the same target distribution to offer probabilistic predictions toward categorizing the medical images to their respective categories. Often, this idealized target is not possible due to the limited data availability, or weak labels. In the present situation, despite a large number of cases worldwide, we have very limited COVID-19 CXR image data that are publicly available to train DL models where the goal is to recognize CXR images showing COVID-19-related viral pneumonia from those caused by other non-COVID-19 viral, bacterial, and other pathogens. Acquiring such data remains a goal for medical societies such as the Radiological Society of North America (RSNA) [7] and Imaging COVID-19 AI Initiative in Europe [8]. The large number of training data enables a diversified feature space across categories that help to enhance inter-class variance, leading to a better DL performance. The absence of such data leads to model overfitting and poor generalization to unseen real-world data. Under these circumstances, data augmentation has been proven to be effective in training discriminative DL models [9]. There are several data augmentation methods discussed in the literature for improving performance in natural computer vision tasks. These include traditional augmentation techniques like flipping, rotations, color jittering, random cropping, and elastic distortions and generative adversarial networks (GAN)-based synthetic data generation [10]. Other methods such as random image cropping and patching (RICAP) [11] are proposed for natural images to augment the training data to achieve superior performance on CIFAR-100 and ImageNet classification tasks.

Unlike natural images, such as those found in ImageNet [12], medical images tend to have different visual characteristics exhibiting high inter-class similarities and highly localized ROI. Under these circumstances, traditional augmentation methods that introduce simple pixel-wise image modifications are shown to be less effective [13]. On the other hand, GAN-based DL models that are used for synthetic data generation are computationally complex and the jury is still out on the anatomical and pathological validity of synthesized images. These networks are hard to train due to the problem of Nash equilibria, defined as the zero-sum game between the generator and the discriminator networks, where they contest with each other in improving performance [14]. Further, these networks are shown to be sensitive to the selection of architecture and hyperparameters and often get into mode collapse, resulting in a degraded performance [14]. In general, there is a great opportunity for research in developing effective data augmentation strategies for medical visual recognition tasks. Goals for such medical data augmentation techniques include reducing overfitting and regularization errors in a data-scarce situation. The urgency offered by the pandemic has led to the motivation behind this study.

In this work, we use weakly labeled CXR images that are automatically pooled from publicly available pneumonia datasets to augment training data toward classifying them into bacterial and viral pneumonia classes and compare the performance with non-augmented training. The goal is to improve COVID-19 detection in CXRs on the hypothesis that it is a kind of viral pneumonia. This would leverage the large collections of images toward meeting an emergent goal.

## 2. Materials and Methods

### 2.1. Data and Workflow

This retrospective analysis was performed using four publicly available CXR collections:

(i) Pediatric CXR dataset [4]: A set of 5232 anterior–posterior (AP) projection CXR images of children of 1 to 5 years of age acquired as part of the routine clinical care at the Guangzhou Children’s Medical Center in China. The set contains 1583 normal, 2780 bacterial pneumonia, and 1493 CXRs showing non-COVID-19 viral pneumonia, respectively;

(ii) RSNA CXR dataset [15]: The RSNA, Society of Thoracic Radiology (STR), and the National Institutes of Health (NIH) jointly organized the Kaggle pneumonia detection challenge to develop image analysis and machine learning algorithms to automatically categorize the CXRs as showing normal, non-pneumonia-related or pneumonia-related opacities. The publicly available data are a curated subset of 26,684 AP and posterior–anterior (PA) CXRs showing normal and abnormal radiographic patterns, taken from the NIH CXR-14 dataset [16]. It includes 6012 CXRs showing pneumonia-related opacities with ground truth (GT) bounding box annotations for these on 1241 CXRs;

(iii) CheXpert CXR dataset [17]: A subset of 4683 CXRs showing pneumonia-related opacities selected from a collection of 223,648 CXRs in frontal and lateral projections, collected from 65,240 patients at Stanford Hospital, California, and labeled for 14 thoracic diseases by extracting the labels from radiological texts using an automated natural language processing (NLP)-based labeler, conforming to the glossary of the Fleischner Society;

(iv) NIH CXR-14 dataset [16]: A subset of 307 CXRs showing pneumonia-related opacities selected from a collection of 112,120 CXRs in frontal projection, collected from 30,805 patients. Images are labeled with 14 thoracic disease labels extracted automatically from radiological reports using an NLP-based labeler;

(v) Twitter COVID-19 CXR dataset: A collection of 135 CXRs showing COVID-19-related viral pneumonia, collected from SARS-CoV-2-positive subjects, has been made available by a cardiothoracic radiologist from Spain via Twitter (https://twitter.com/ChestImaging). The images are made available in JFIF format at approximately a 2K × 2K resolution;

(vi) Montreal COVID-19 CXR dataset: As of 14 April 2020, a collection of 179 SARS-CoV-2-positive CXRs and others showing non-COVID-19 viral disease manifestations has been made publicly available by the authors of [18] in their GitHub repository. The CXRs are made available in AP and PA projections.

Table 1, Table 2 and Table 3 show the distribution of the data used toward the baseline training and evaluation, weak-label augmentation, and COVID-19 classification, respectively. The GT disease bounding box annotations for a sample of the COVID-19 CXR data, containing 27 CXRs collectively from the Twitter COVID-19 and Montreal COVID-19 CXR collections, were set by the verification of publicly identified cases from an expert radiologist who annotated the sample test collection.

Figure 2 illustrates the graphical abstract of the proposed study. Broadly, our workflow consisted of the following steps: First, we preprocessed the images to make them suitable for use in DL. Then, as shown in Figure 2a, we evaluated the performance of a custom CNN and a selection of pre-trained CNN models for categorizing the pediatric CXR collection, referred to as baseline, into bacterial or viral pneumonia. The trained model was further evaluated for its ability to categorize the publicly available COVID-19 CXR collections as showing viral pneumonia. Next, as shown in Figure 2b, we used the trained model from Figure 2a to weakly label CXRs as showing bacterial or viral pneumonia in other pneumonia datasets (RSNA, CheXpert, and NIH). Then, as shown in Figure 2c, the baseline training data were augmented with these weakly labeled CXRs to improve the detection performance with both (i) the baseline test data and (ii) the COVID-19 CXR collections.

This discriminative training data augmentation strategy recognizes biological similarity in viral and COVID-19 pneumonia, i.e., both are viral; however, it also notes the distinct radiological manifestations between each other as well as with non-viral pneumonia-related opacities. Rejects from the classifier developed in this study are not necessarily normal and should be subjected to a separate clinical assessment.

### 2.2. Lung ROI Segmentation and Preprocessing

It is important to add controls during the training of the data-driven DL methods for disease screening/diagnosis. Learning irrelevant feature representations could adversely impact the clinical decision-making. To assist the DL model to focus on pulmonary abnormalities, we used a dilated dropout U-Net [19] to segment the lung ROI from the background. Dilated convolutions are shown to improve performance [20] with exponential receptive field expansion while preserving spatial resolution with no added computational complexity. A Gaussian dropout with an empirically determined value of 0.2 was used after the convolutional layers in the network encoder to avoid overfitting and improve generalization. A publicly available collection of CXRs and their associated lung masks [21] was used to train the dilated dropout U-Net model to generate lung masks of 224 × 224 pixel resolution. Callbacks were used to store the best model weights after each epoch. The generated masks were superimposed on the original CXRs to delineate the lung boundaries, crop them to the size of a bounding box, and re-scale them to 224 × 224 pixel resolution to reduce the computational complexity. Figure 3 shows the segmentation steps performed in this study.

Additional preprocessing steps performed were as follows: (i) CXRs were thresholded at to remove very bright pixels to remove text annotations (empirically determined to be in the range (235–255) that might be present in the cropped images. Missing pixels were in-painted using the surrounding pixel values. (ii) Images were normalized to make the pixel values lie in the range (0–1). (iii) CXR images were median-filtered to remove noise and preserve edges. (iv) Image pixel values were centered and standardized to reduce the computational complexity. Next, the cropped CXRs were used to train and evaluate a custom CNN and a selection of pretrained models at the different learning stages performed in this study.

### 2.3. Models and Computational Resources

The performance of a custom CNN model whose design is inspired by the wide residual network (WRN) architecture proposed in [22] and a selection of ImageNet pretrained CNN models was evaluated during the different stages of learning performed in this study. The benefit of using a WRN compared with the traditional residual networks (ResNets) [23] is that it is shallower, resulting in shorter training times while producing similar or improved accuracy. In this study, we used a WRN-based custom CNN architecture with dropouts used in every residual block. After the pilot empirical evaluations, we used a network depth of 28, a width of 10, and a dropout ratio of 0.3 for the custom WRN used in this study.

We evaluated the performance of the following pretrained CNN models, viz., (a) VGG-16 [24], (b) Inception-V3 [25], (c) Xception [26], (d) DenseNet-121 [27], and (e) NasNet-mobile [28]. The pretrained CNNs were instantiated with their ImageNet [12] pretrained weights and truncated at their fully connected layers. The output feature maps were global average-pooled and fed to a final dense layer with Softmax activations to output the prediction probabilities.

The following hyperparameters of the custom WRN and pretrained CNNs were optimized through a randomized grid search method: (i) momentum, (ii) L2-weight decay, and (iii) initial learning rate of the stochastic gradient descent (SGD) optimizer. We initialized the search ranges to (0.80–0.99), (1 × 10^−8^–1 × 10^−2^), and (1 × 10^−7^–1 × 10^−3^) for the learning momentum, L2-weight decay, and initial learning rate, respectively. The custom WRN was initialized with random weights and the pretrained models were fine-tuned end-to-end with smaller weight updates to make them data-specific and classify the CXRs to their respective categories. Callbacks were used to monitor the model performance with the validation data and store the best model weights for further analysis with the hold-out test data.

The performances of the custom WRN and the pretrained CNN models were evaluated in terms of (i) accuracy, (ii) area under the curve (AUC), (iii) sensitivity or recall, (iv) specificity, (v) precision, (vi) F-score, and (vii) Mathews correlation coefficient (MCC). The models were trained and evaluated on a Windows System with Intel Xeon CPU 3.80 GHz with 32 GB RAM and NVIDIA GeForce GTX 1070 GPU. We used Keras 2.2.4 API version with Tensorflow backend and CUDA/CUDNN dependencies.

### 2.4. Weakly Labeled Data Augmentation

Our approach builds from following the literature which stated that CXRs showing COVID-19 viral pneumonia manifestations are visually similar to, yet distinct from, those caused by bacterial, fungal, and other non-COVID-19-related viral pneumonia [2]. First, we trained the custom WRN and the pretrained models on the pediatric CXR collection [4] and evaluated them on the ability to categorize the hold-out test data, listed in Table 1, into bacterial or viral pneumonia types. We selected the best performing model on this baseline data. Next, we conducted two evaluations with this model: (i) we identified viral pneumonia CXRs from the Twitter-COVID-19 and Montreal-COVID-19 collections; and (ii) we evaluated its performance in weakly categorized CXRs showing pneumonia of an unknown type from the RSNA, CheXpert, and NIH CXR collections, listed in Table 2, as belonging to the bacterial or viral pneumonia opacity categories. These weakly classified CXRs were used to augment the baseline training data. This weakly labeled augmentation was motivated by the need to expand the learned feature space. The augmentation enabled the following: (i) to make the training distribution encompass the variability in the test distribution; (ii) to enhance the inter-class discrimination; and (iii) to decrease the generalization error by training with samples from a diversified distribution. The model was trained with various combinations of the augmented training data and evaluated against the baseline test data and CXRs identifying viral pneumonia from the Twitter-COVID-19 and Montreal-COVID-19 CXR collections.

### 2.5. Salient ROI Localization

Visualization helps in interpreting the model predictions and identify the salient ROI involved in decision-making. In this study, the learned behavior of the best performing baseline model in categorizing the CXRs to the bacterial and viral pneumonia classes was visualized through gradient-weighted class activation maps (Grad-CAM) [29]. Grad-CAM is a gradient-based visualization method where the gradients for a given class are computed concerning the features extracted from the deepest convolutional layer in a trained model and are fed to a global average pooling layer to obtain the weights of importance involved in decision-making. This results in a two-dimensional heat map which is a weighted combination of the feature maps involved in categorizing the image to its respective class.

## 3. Results

Table 4 shows the optimal hyperparameter values obtained using a randomized grid search for the custom WRN and pretrained CNNs. These are used for the model training and evaluation. For the model validation, we allocated 20% of the training data which was randomly selected. The performance achieved by the models is shown in Table 5.

It can be observed that the VGG-16 model demonstrates superior performance in terms of accuracy and AUC with the baseline test data. The Xception model gives higher precision and specificity than the other models. However, the VGG-16 model outperformed the others in classifying the pediatric CXRs as showing bacterial or viral pneumonia when considering the F-score and MCC. Both these scores provide a balanced precision and sensitivity measure. The performance excellence of the VGG-16 model can be attributed to (i) the optimal architecture depth for learning the data and (ii) the ability to extract diversified features that categorize the CXRs to their respective categories. These deductions are supported by the reduced performance of deeper models like DenseNet-121 which possibly suffered from overfitting. Therefore, we select the VGG-16 model for further evaluating against the Twitter-COVID-19 and Montreal-COVID-19 CXR collections as showing viral pneumonia. The performance achieved is shown in Table 6. Figure 4 shows the confusion matrix obtained toward classifying the Twitter- and Montreal-COVID-19 CXR collections as showing viral pneumonia.

It was surprising to observe, from Table 6 and Figure 4, that the baseline-trained VGG-16 model did not deliver superior performance in identifying COVID-19 CXRs in the Twitter- and Montreal-COVID-19 CXR collections. We attribute this to two possibilities: (i) limited variance in the training distribution and hence a narrow feature space to learn the related patterns; or (ii) that COVID-19 manifestation is distinct from viral pneumonia even though it is caused by the SARS-CoV-2 virus.

The learned behavior of the baseline-trained VGG-16 model with the pediatric CXR and COVID-19 CXR collections is interpreted through Grad-CAM visualizations and is shown in Figure 5.

The gradients for the bacterial and viral pneumonia classes that are flowing into the deepest convolutional layer of the trained model are used to interpret the neurons involved in the decision-making. The heat maps obtained as a result of weighing these feature maps are superimposed on the original CXRs to identify the salient ROI involved in categorizing the CXRs to their respective classes. It is observed that the model is correctly focusing on the salient ROI for the baseline test data coming from the same training distribution that helps to categorize them into bacterial and viral pneumonia classes. However, the salient ROI involved in categorizing an image from the Montreal-COVID-19 CXR collection that comes from a different distribution compared with the baseline data did not properly overlap with the GT annotations. This further underscores the inference above that the model did not learn the disease manifestations in the aforementioned COVID-19 CXR collections, suggesting that their appearances are distinct.

With data-driven DL methods, the training data may contain samples that do not contribute to decision-making. Modifying the training distribution could provide an active solution to improve performance with a similar and/or different test distribution. In response, our approach is to expand the training data feature space to create a diversified distribution that could help learn and improve the performance with the baseline test data coming from the same distribution as the training data and/or with other test data coming from a different distribution. In this study, we propose to expand the training data feature spaces by augmenting them with weakly classified CXR images. For this, the best-performing, baseline-trained VGG-16 model is used to weakly classify the CXR images from the NIH, RSNA, and CheXpert collections showing pneumonia-related opacities as showing bacterial or viral pneumonia. The weakly labeled images are further used to augment the baseline training data to evaluate for an improvement in performance toward categorizing the pediatric CXR test, Twitter-COVID-19, and Montreal-COVID-19 CXR collections. Table 7 shows the number of samples across the bacterial and viral pneumonia categories after augmenting the baseline pediatric CXR training data with weakly labeled images from the respective CXR collections. The performance metrics achieved with the augmented training data are shown in Table 8.

Note that the baseline training data augmented with weakly labeled CXR images from the CheXpert CXR collection demonstrated superior performance in all metrics compared with the non-augmented and other training data augmentations. This underscores the fact that this augmentation approach resulted in a favorable increase in the training data size, encompassing a diversified distribution to learn and improve the performance in the baseline test data, compared with that of the non-augmented training. We studied the effect of weakly labeled data augmentation in classifying the Twitter- and Montreal-COVID-19 CXR collections as belonging to the viral pneumonia category. The results are as shown in Table 9.

The empirical evaluations demonstrate that the baseline training data augmented with the weakly labeled CXR images from the CheXpert collection improved the performance with an accuracy of 0.5555 and 0.6536, as compared with the non-augmented baseline (0.2885 and 0.5028) in classifying the Twitter- and Montreal-COVID-19 CXR collection, respectively, as belonging to the viral pneumonia category. The performance degradation with other combinations of weakly labeled data augmentation underscores the fact that (i) adding more data introduces noise into the training process and (ii) increasing the number of training samples does not always improve performance.

## 4. Discussion

In this section, we present the results from our analyses following our suspicion that even though COVID-19 pneumonia is caused by a virus (SARS-CoV-2), its manifestations in CXRs are distinct from other viral pneumonia patterns. To test our hypothesis, we introduced the Twitter- and Montreal-COVID-19 CXR collections, separately, to the best-performing weakly labeled augmented training data, i.e., Baseline + CheXpert. This is illustrated in Figure 6, and the results are shown in Table 10, below.

The results in Table 10 support our hypothesis that augmenting the best-performing weakly labeled augmented training data with class-specific data, since they are sufficiently distinct, is necessary to obtain improvement. We are intrigued by the disparity in improvement, however. Recall that the Twitter-COVID-19 collection was posted from a hospital in Spain. In contrast, the Montreal-COVID-19 collection is sourced broadly and does not typify the pneumonia opacity from a select population. Thus, the variety introduced by augmenting with the Montreal-COVID-19 data results in a much greater boost in performance as compared with Twitter-COVID-19.

Next, to test the degree to which COVID-19 is distinct from routine viral pneumonia manifestations, we augmented the baseline directly with the individual COVID-19 images.

It is observed from Table 11 that augmenting the baseline training data with the Twitter-COVID-19 CXR collection significantly improved the performance in detecting COVID-19 CXRs in the Montreal collection as belonging to the viral pneumonia category. We observed similar improvements in performance with the Twitter-COVID-19 CXRs when the baseline training data is augmented with the Montreal-COVID-19 CXR collection. This suggests that weakly labeled augmentation might be hurting rather than helping the detection of COVID-19. While this may seem counter to our original hypothesis, recall that weakly labeled augmentation is very valuable when there are insufficient data for a subclass. This is supported by the results shown in Table 8 above. In the case of COVID-19, note that the collections are very small and need some additional training images. Therefore, these augmented training images must be selected wisely.

Confusion matrices for the results in Table 11 above are shown in Figure 7, while Figure 8 shows the learned behavior of the trained model. We observe that the learned interpretation is correctly focusing on the salient ROI, matching with the GT annotations that help to categorize COVID-19 CXRs as showing viral pneumonia. This is a significant improvement over the non-augmented training results shown in Figure 5.

## 5. Conclusions and Future Work

Weakly labeled data augmentation helped to improve performance with the baseline test data because the CXRs with pneumonia-related opacities in the CheXpert collection have a similar distribution to bacterial and non-COVID-19 viral pneumonia. This similarity helped to expand the training feature space by introducing a controlled class-specific feature variance that improves performance with the baseline test data. However, with COVID-19 CXRs, weakly labeled data augmentation did not deliver superior performance on its own, primarily due to the small data set size—which is the base reason for weakly labeled augmentation with data from other collections—and distinct opacity patterns compared with other viral and bacterial pneumonia. In clinical use, it could quickly help to separate patients with COVID-19 opacities (true positives) and refer the rest for further clinical assessment. As future work, we aim to expand the analysis to multi-class problems. Constructing model ensembles to combine the predictions of models trained on various combinations of augmented training data might further improve the COVID-19 detection performance.

## Figures and Tables

**Figure 1 diagnostics-10-00358-f001:**
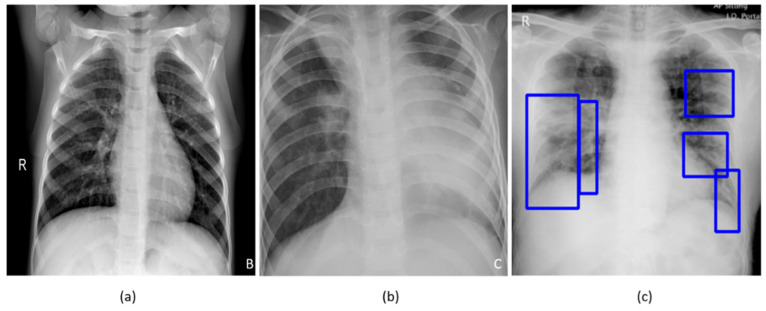
Chest X-rays (CXRs) showing (**a**) clear lungs; (**b**) bacterial pneumonia infection manifesting as consolidations in the right upper lobe and retro-cardiac left lower lobe; and (**c**) COVID-19 pneumonia infection showing bilateral manifestations. Blue frames in (**c**) denote radiologist annotations indicating disease regions, which serve as ground truth in our analysis.

**Figure 2 diagnostics-10-00358-f002:**
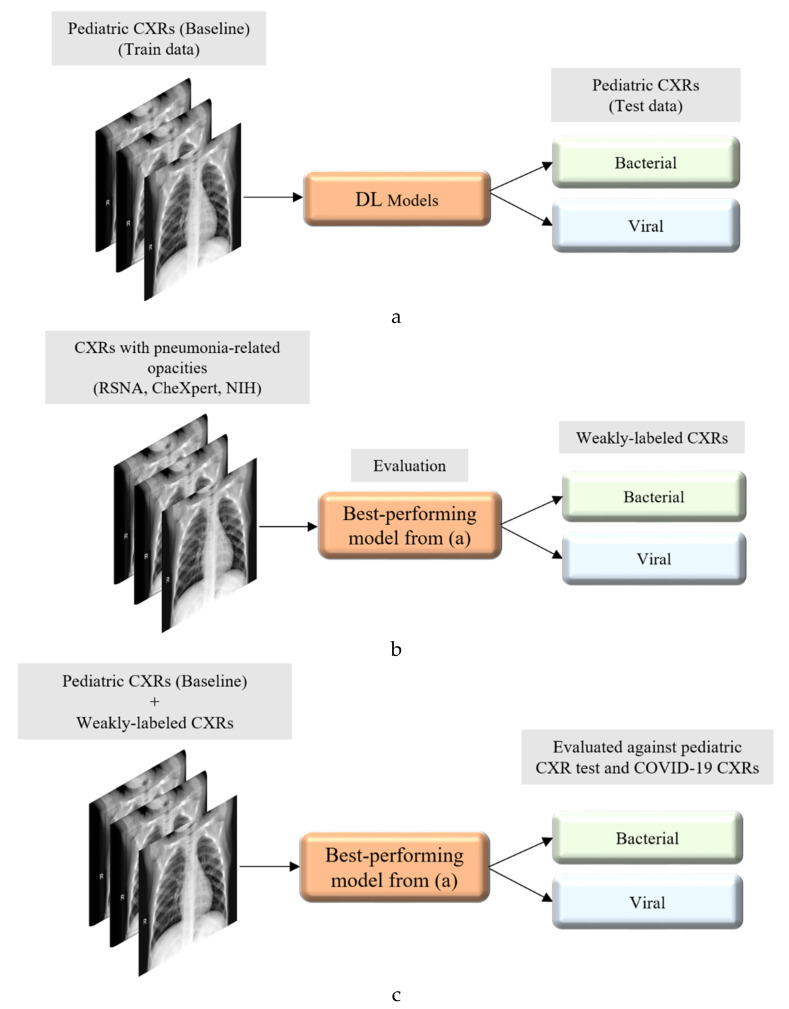
Graphical abstract of the proposed study. (**a**) Model training and evaluation with baseline pediatric CXR data; (**b**) using the best performing model from (**a**) to weakly classify CXRs from Radiological Society of North America (RSNA), National Institutes of Health (NIH), and CheXpert containing pneumonia-related opacities, as showing bacterial or viral pneumonia; and (**c**) augmenting the baseline training data with weakly labeled CXRs to check for performance improvement.

**Figure 3 diagnostics-10-00358-f003:**
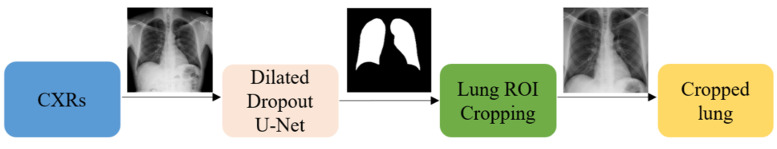
The segmentation approach showing dilated dropout U-Net-based mask generation and lung ROI cropping.

**Figure 4 diagnostics-10-00358-f004:**
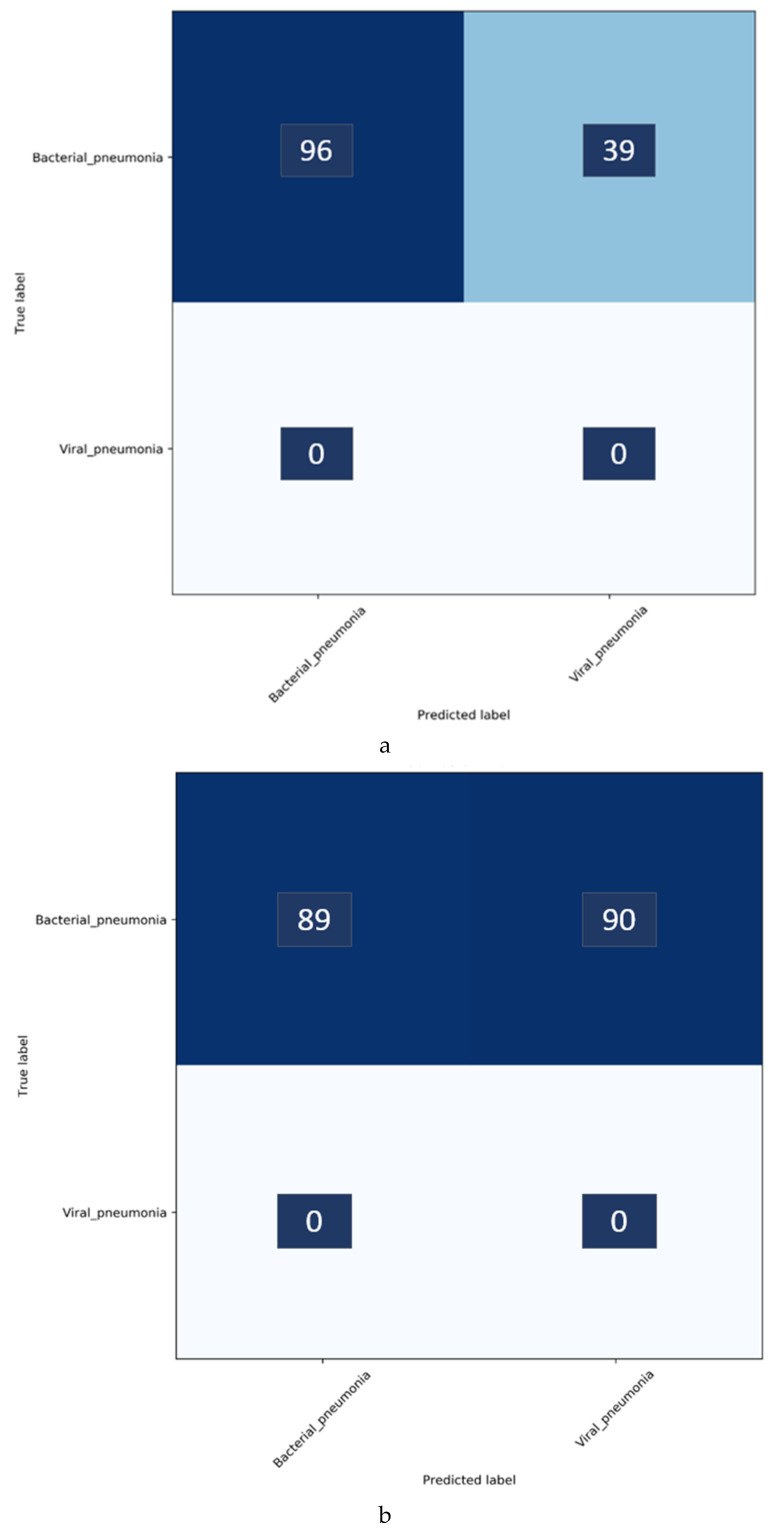
Confusion matrix after classifying bacterial and viral pneumonia in the (**a**) Twitter-COVID-19 and (**b**) Montreal-COVID-19 CXR collections. Enlarged text labels have been manually superimposed for clarity.

**Figure 5 diagnostics-10-00358-f005:**
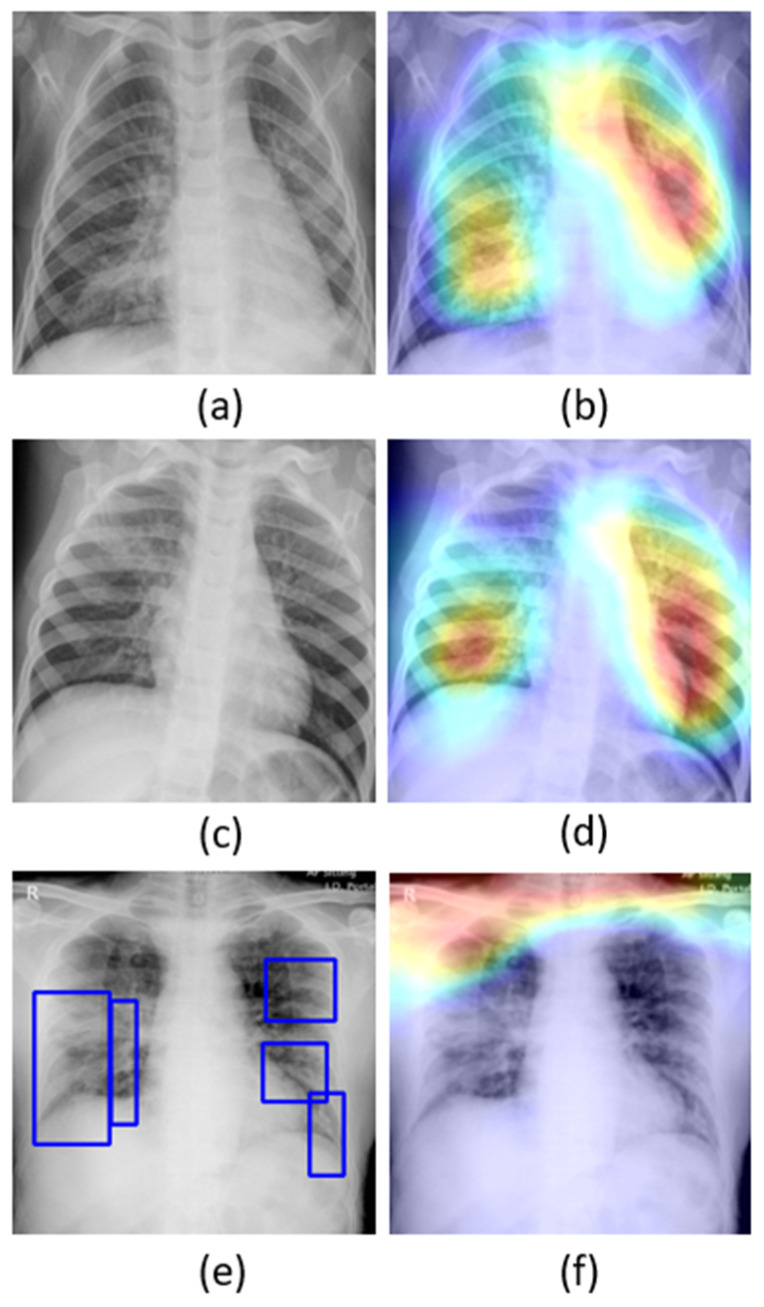
Original CXRs and their salient ROI visualization: (**a**,**b**) show a CXR with bilateral bacterial pneumonia and the corresponding Grad-CAM visualization; (**c**,**d**) show a CXR with viral pneumonia manifestations and the corresponding salient ROI visualization; and (**e**,**f**) show a sample CXR from the Montreal-COVID-19 CXR collection with ground truth (GT) annotations and corresponding salient ROI visualization. Blue frames in (**e**) denote radiologist annotations indicating disease regions, which serve as ground truth in our analysis.

**Figure 6 diagnostics-10-00358-f006:**
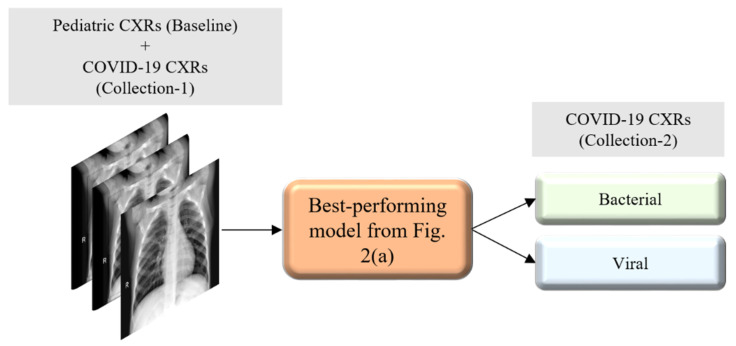
Evaluating the performance against a collection of COVID-19 CXRs when augmenting the best-performing weakly labeled augmented training data with a different collection of COVID-19 CXRs.

**Figure 7 diagnostics-10-00358-f007:**
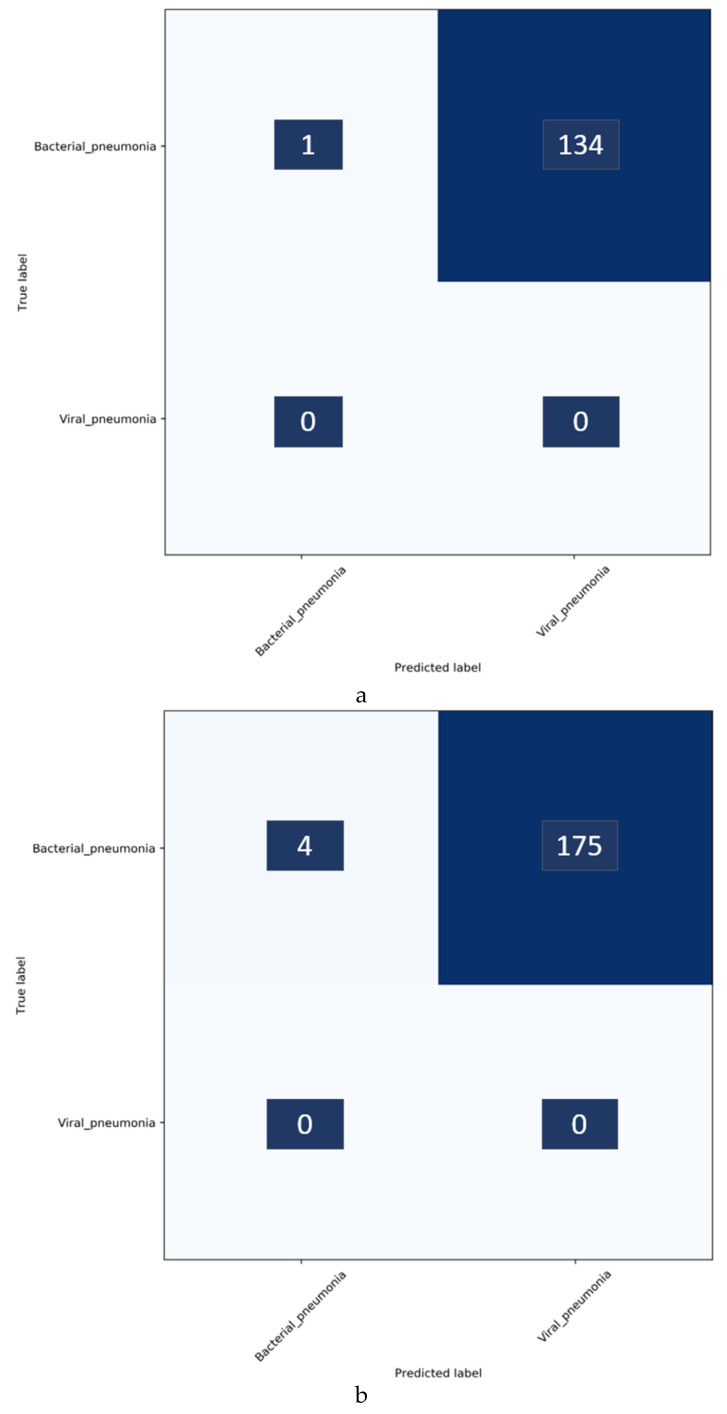
Confusion matrix after classifying bacterial and viral pneumonia in the (**a**) Twitter- and (**b**) Montreal-COVID-19 CXR collections after augmenting the baseline training data with individual COVID-19 CXR collections. Enlarged text labels have been manually superimposed for clarity.

**Figure 8 diagnostics-10-00358-f008:**
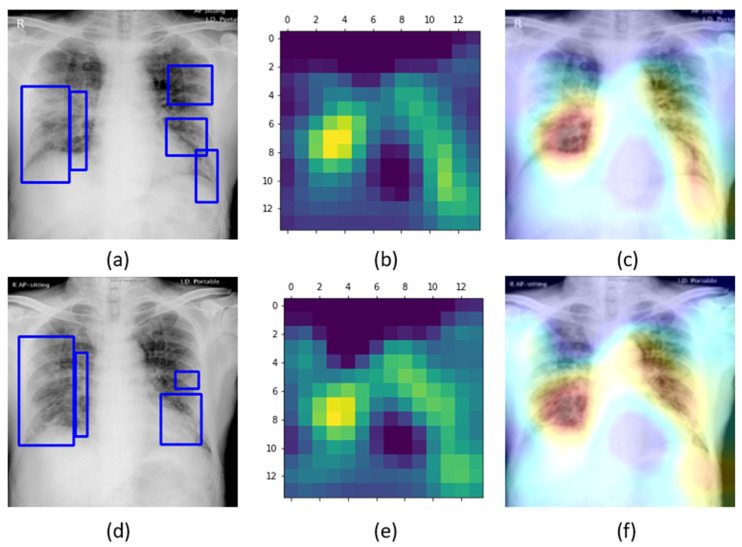
Original CXRs, heat maps, and salient ROI visualization: (**a**–**c**) show a sample Montreal-COVID-19 CXR with GT annotations, the corresponding heat map, and Grad-CAM visualization; (**d**–**f**) show a sample Twitter-COVID-19 CXR with GT annotations, the heat map, and its associated class activation maps. Blue frames in (**a**,**d**) denote radiologist annotations indicating disease regions, which serve as ground truth in our analysis.

**Table 1 diagnostics-10-00358-t001:** Baseline dataset characteristics. Numerator and denominator denote the number of train and test data, respectively. Note that this dataset predates the onset of SARS-CoV2 virus, and therefore the viral pneumonia is of non-COVID-19 type.

Dataset	Bacterial (Proven) Pneumonia	Viral (Proven) Pneumonia
Pediatric	2538/242	1345/148

**Table 2 diagnostics-10-00358-t002:** Characteristics of datasets used for weak-label classification.

Dataset	Pneumonia of Unknown Type
RSNA	6012
CheXpert	4683
NIH	307

**Table 3 diagnostics-10-00358-t003:** Distribution of COVID-19 CXR data.

Dataset	COVID-19 Viral Pneumonia
Twitter COVID-19	135
Montreal COVID-19	179

**Table 4 diagnostics-10-00358-t004:** Optimal values for the hyperparameters for the custom wide residual network (WRN) and pretrained convolutional neural networks (CNNs) obtained through the randomized grid search (M: momentum, ILR: initial learning rate, and L2: L2-weight decay).

Models	Optimal Values		
	M	ILR	L2
Custom	0.90	1 × 10^−3^	1 × 10^−5^
Pretrained	0.95	1 × 10^−3^	1 × 10^−6^

**Table 5 diagnostics-10-00358-t005:** Performance achieved by the deep learning (DL) models in classifying the pediatric CXR dataset (baseline) into bacterial and viral categories. Here, Acc.: accuracy, Sens.: sensitivity, Prec.: precision, F: F-score, and MCC: Matthews correlation coefficient.

Models	Acc.	AUC	Sens.	Spec.	Prec.	F	MCC
Custom WRN	0.8974	0.9534	0.9381	0.8311	0.9008	0.9191	0.7806
VGG-16	**0.9308**	**0.9565**	0.9711	0.8649	0.9216	**0.9457**	**0.8527**
Inception-V3	0.9103	0.937	0.9587	0.8311	0.9028	0.9299	0.8085
Xception	0.9282	0.954	0.9546	**0.8852**	**0.9315**	0.9429	0.8469
DenseNet-121	0.9026	0.9408	0.967	0.7973	0.8864	0.925	0.7931
NASNet-mobile	0.9282	0.9479	**0.9753**	0.8514	0.9148	0.944	0.8477

Bold numerical values denote superior performance.

**Table 6 diagnostics-10-00358-t006:** Performance metrics achieved in classifying the Twitter- and Montreal-COVID-19 CXR collections as showing viral pneumonia.

Model	Accuracy	
	Twitter-COVID-19	Montreal-COVID-19
VGG-16	0.2885	0.5028

**Table 7 diagnostics-10-00358-t007:** Number of samples in weakly labeled augmented training data.

Dataset	BP	VP
Baseline + NIH	2720	1470
Baseline + CheXpert	4683	3883
Baseline + RSNA	6577	3318
Baseline + NIH + CheXpert	4865	4008
Baseline + NIH + RSNA	6759	3443
Baseline + CheXpert + RSNA	8722	5856
Baseline + NIH + CheXpert + RSNA	8904	5981

**Table 8 diagnostics-10-00358-t008:** Performance metrics achieved with the different combinations of augmented training data toward classifying the pediatric CXR (baseline) test data into bacterial and viral pneumonia categories.

Dataset	Acc.	AUC	Sens.	Spec.	Prec.	F	MCC
Baseline	0.9308	0.9565	0.9711	0.8649	0.9216	0.9457	0.8527
Data augmentation with weakly labeled images							
Baseline + NIH	0.9179	0.9600	0.9587	0.8514	0.9134	0.9355	0.8249
Baseline + CheXpert	**0.9405**	**0.9689**	**0.9877**	**0.8624**	**0.9201**	**0.9542**	**0.8716**
Baseline + RSNA	0.9359	0.9592	**0.9877**	0.8514	0.9158	0.9503	0.8653
Baseline + NIH + CheXpert	0.9333	0.9606	0.9835	0.8514	0.9154	0.9483	0.8594
Baseline + NIH + RSNA	0.9231	0.9642	0.9959	0.8041	0.8926	0.9415	0.8411
Baseline + CheXpert + RSNA	0.9359	0.9628	0.9835	0.8582	0.919	0.9501	0.8647
Baseline + NIH + CheXpert + RSNA	0.9154	0.9542	0.9794	0.8109	0.8944	0.935	0.8217

Bold numerical values denote superior performance.

**Table 9 diagnostics-10-00358-t009:** Performance metrics achieved through weakly labeled data augmentation toward classifying the Twitter- and Montreal-COVID-19 CXR collections as belonging to the viral pneumonia category.

Dataset	Accuracy	
	Twitter-COVID-19	Montreal-COVID-19
Baseline	0.2885	0.5028
Data augmentation with weakly labeled images		
Baseline + NIH	0.1037	0.2625
Baseline + CheXpert	**0.5555**	**0.6536**
Baseline + RSNA	0.2296	0.4469
Baseline + NIH + CheXpert	0.1852	0.4078
Baseline + NIH + RSNA	0.1407	0.4413
Baseline + CheXpert + RSNA	0.2222	0.4357
Baseline + NIH + CheXpert + RSNA	0.1852	0.4413

Bold numerical values denote superior performance.

**Table 10 diagnostics-10-00358-t010:** Performance metrics achieved using augmenting the best-performing weakly labeled augmented training data with one of the COVID-19 CXR collections toward classifying another COVID-19 CXR collection as belonging to the viral pneumonia category.

Dataset	Accuracy	
	Twitter-COVID-19	Montreal-COVID-19
Baseline	0.2885	0.5028
Baseline + CheXpert	0.5555	0.6536
Baseline + CheXpert + Twitter	-	**0.7095**
Baseline + CheXpert + Montreal	**0.8889**	-

Bold numerical values denote superior performance.

**Table 11 diagnostics-10-00358-t011:** Performance metrics achieved using augmenting training data directly with one of the COVID-19 CXR collections toward classifying another COVID-19 CXR collection as belonging to the viral pneumonia category.

Dataset	Accuracy	
	Twitter-COVID-19	Montreal-COVID-19
Baseline	0.2885	0.5028
Baseline + Twitter-COVID-19	-	**0.9778**
Baseline + Montreal-COVID-19	**0.9926**	-

Bold numerical values denote superior performance.

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
