# Peer review of "Weakly Labeled Data Augmentation for Deep Learning: A Study on COVID-19 Detection in Chest X-Rays"

_diagnostics, 2020, doi:10.3390/diagnostics10060358_

Round 1

Reviewer 1 Report

In my opinion, this paper is difficult to read. Maybe, two-stage training of deep learning model was used in this study. In the first stage, the dataset A of Table 1 was used. Because the training data and the test data in the second stage might be variable, this confuses me. In addition, definition of weak label is unclear for me.

Major points

“We  also  evaluated  the  performance  of  the  best  performing  baseline  model  in  weakly  categorizing the CXRs showing pneumonia-related opacities from RSNA, CheXpert, and NIH CXR  collections as belonging to the bacterial or viral pneumonia categories.” According to Table 1, the dataset B, C, and D has only UP. What is weak label for UP? How do authors use UP cases as weak label for data augmentation? The detail of these two points are not described.  

Ref. 6 is published as preprint. The paper is published at April 2020. Therefore, in my opinion, the preprint paper is difficult to use as ref. at the current time.

What is “Baseline” in Table 5? Authors mean the A dataset?  

In Table 6, the rows of Baseline + Twitter and Baseline + Montreal should be deleted, because data augmentation with weakly labeled images was not used in these two.

This study built the model which classified pneumonia. However, it seems that the model ignored normal cases.

This study used train/test splitting. Compared with train/validation/test splitting or 10-fold cross validation, train/test splitting may cause overfit.

Minor points

“Callbacks are used to monitor model performance and store the best model weights for further analysis.” This may cause overfit if authors monitored performance in test data of COIVT-19.   

“On the other hand, GAN-based DL models that are used  for  synthetic  data  generation  are  computationally  complex  and  the  jury  is  still  out  on  the  anatomical and pathological validity of synthesized images.” Recently-proposed data augmentation methods should be referred to. Such as, mixup, random erasing, ricap, and so on. In addition, papers using these recently-proposed data augmentation methods and medical images should be cited.

“These  include  traditional  augmentation  techniques  like  flipping, rotations,  color  jittering,  random  cropping,  and  elastic  distortions  and generative  adversarial networks (GAN) based synthetic data generation [8].”  Ref. 8 is not related with medical images. Please cite papers where GAN was used for data augmentation of medical images.

Author Response

Q1: In my opinion, this paper is difficult to read. Maybe, two-stage training of deep learning model was used in this study. In the first stage, the dataset A of Table 1 was used. Because the training data and the test data in the second stage might be variable, this confuses me. In addition, definition of weak label is unclear for me.

Author response:  We render our sincere thanks to the reviewer for his valuable time and comments toward our study. We regret the lack of clarity in our initial submission. In the revised manuscript, we have modified the way we address the use of different datasets to ensure readability. We have also included a discussion on how the CXRs are weakly labeled. We revised the Tables and their associated captions to convey clarity. The abstract is revised to include experimental results.

Q2. “We  also  evaluated  the  performance  of  the  best  performing  baseline  model  in  weakly  categorizing the CXRs showing pneumonia-related opacities from RSNA, CheXpert, and NIH CXR  collections as belonging to the bacterial or viral pneumonia categories.” According to Table 1, the dataset B, C, and D has only UP. What is weak label for UP? How do authors use UP cases as weak label for data augmentation? The detail of these two points are not described.   

Author response:  We appreciate the reviewer’s concerns in this regard. We regret the lack of clarity in addressing the use of different datasets in this study. In the revised manuscript, we have made changes to Table 1 as shown below:

Table 1. Dataset characteristics. Numerator and denominator denote the number of train and test data respectively (UP=Pneumonia of unknown type, BP= Bacterial (proven) pneumonia, VP= non-COVID-19 viral (proven) pneumonia, CP = COVID-19 pneumonia).

Dataset

UP

BP

VP

CP

Pediatric

-

2538/242

1345/148

-

RSNA

-/6012

-

-

-

CheXpert

-/4683

-

-

-

NIH

-/307

-

-

-

Twitter COVID-19

-

-

-

-/135

Montreal COVID-19

-

-

-

-/179

BP and VP refer to the proven bacterial and non-COVID-19 viral pneumonia CXRs that come from the Pediatric CXR dataset. UP refers to the unknown pneumonia CXRs that come from RSNA, CheXpert, and NIH CXR-14 collection. CP refers to the COVID-19 viral pneumonia that comes from Twitter and Montreal COVID-19 collection.

Broadly, our workflow consists of the following steps:

First, we preprocess the images to crop them to the size of a bounding box containing only the lung pixels to make them suitable for use in DL and learn only relevant features toward classification.

Then, we evaluate the performance of a custom CNN and a selection of pre-trained CNN models for binary categorization of the pediatric CXR collection as showing bacterial or viral pneumonia.

Next, we use the trained model to weakly label the CXRs in RSNA, CheXpert, and NIH collections with unknown pneumonia-related opacities (UP) as showing bacterial or non-COVID-19 viral pneumonia.

We use the trained model to also categorize the publicly available COVID-19 CXR collections as showing viral pneumonia.

Then, the baseline non-augmented training data from pediatric CXR collection is augmented with these weakly labeled CXRs from RSNA, ChexPert, and NIH collections to improve detection performance with the baseline hold-out test data and the COVID-19 CXR collections.

We also augment the non-augmented training data from pediatric CXR collection with COVID-19 CXRs from one of the two different collections (Twitter and Montreal) to evaluate for an improvement in performance in detecting CXRs showing COVID-19 viral pneumonia from the other collection.

Through empirical evaluations, we observed that the strategy reduces the intra-class similarity and enhances inter-class discrimination in the strategic ordering of the coarsely labeled data. We have addressed these changes in the revised manuscript.

We have also included a graphical abstract of the proposed study (Fig. 2) in the revised manuscript.

Q3. Ref. 6 is published as preprint. The paper is published at April 2020. Therefore, in my opinion, the preprint paper is difficult to use as ref. at the current time.

Author response:  We sincerely thank the reviewer for his insightful comments. The discriminative training data augmentation strategy proposed in this study recognizes the biological similarity in viral pneumonia and radiological manifestation due to COVID-19 caused respiratory disease and dissimilarity to bacterial and non-COVID-19 viral pneumonia-related opacities. In this work, we show that it is possible to obtain a classifier that uses a collection of weakly-labeled image datasets that are related to, but clinically different from, a subclass (i.e., COVID-19) and identify it with high precision. Conversely, our prior work proposes an iteratively pruned deep learning ensemble strategy [6] that separates normal CXRs from those with COVID-19 and other pneumonia-like opacities. We have cited other works [2] [3] and [5] that discuss the use of CTs and CXRs for COVID-19 classification. Most other efforts are still in the preprint stage.

Q4. What is “Baseline” in Table 5? Authors mean the A dataset? 

Author response:  We regret the lack of clarity. It refers to the non-augmented training data of the pediatric CXR dataset. We evaluate the baseline performance of a custom CNN and a selection of pre-trained CNN models for binary categorization of the baseline, non-augmented pediatric CXR collection as showing bacterial or viral pneumonia. In the revised manuscript, the datasets are referred to, by their respective names in the Tables and within the text.

Q5. In Table 6, the rows of Baseline + Twitter and Baseline + Montreal should be deleted, because data augmentation with weakly labeled images was not used in these two.

Author response:  We render our sincere thanks to the reviewer for his valuable comments in this regard. We have modified the Table and the associated captions to convey clarity. Baseline refers to the pediatric CXR non-augmented training data. This information is added to the Table captions as shown below:

Table 6. Performance metrics achieved using combinations of the augmented training data toward classifying Twitter and Montreal COVID-19 CXR collections as belonging to the viral pneumonia category. Baseline refers to the pediatric CXR non-augmented training data. Bold values indicate superior performance.

Dataset

Accuracy

Twitter-COVID-19

Montreal-COVID-19

Baseline

0.2885

0.5028

Data augmentation with weakly labeled images

Baseline + NIH

0.1037

0.2625

Baseline + CheXpert

0.5555

0.6536

Baseline + RSNA

0.2296

0.4469

Baseline + NIH + CheXpert

0.1852

0.4078

Baseline + NIH + RSNA

0.1407

0.4413

Baseline + CheXpert + RSNA

0.2222

0.4357

Baseline + NIH + CheXpert + RSNA

0.1852

0.4413

*Bold values denote superior performance.

Table 7. Performance metrics achieved using augmenting training data with one of COVID-19 CXR collections toward classifying another COVID-19 CXR collection as belonging to the viral pneumonia category. Baseline refers to the pediatric CXR non-augmented training data.

Dataset

Accuracy

Twitter-COVID-19

Montreal-COVID-19

Baseline

0.2885

0.5028

Baseline + Twitter COVID-19

-

0.9778

Baseline + Montreal COVID-19

0.9926

-

*Bold values denote superior performance.

The discussion on empirical evaluations, as reported in Table 6 and Table 7 are included in lines 362-393.

Q6. This study built the model which classified pneumonia. However, it seems that the model ignored normal cases.

Author response:  We appreciate the reviewer’s concern in this regard. The method proposed in this manuscript is not intended to be viewed as a system for screening patients. Rather, as the title suggests it is a study in the use of weakly labeled sets to devise a highly selective classifier (identifying capability) of a particular class or subclass. COVID-19 provided this unique opportunity and emergent need to test out our hypothesis. The hypothesis is that if we have a large collection of weakly labeled image datasets that are related to but clinically different from a subclass, it is possible to develop a classifier that identifies such images with high precision. Rejects from this classifier are not necessarily normal and should be subjected to separate clinical assessment. In this case, COVID-19 is a virus that causes pulmonary opacities, which are viral-pneumonia like yet distinct from them, and certainly different from bacterial or fungal infections causing similar pulmonary pneumonia. This idea has been mentioned in lines 162-176 in the manuscript.

Q7. This study used train/test splitting. Compared with train/validation/test splitting or 10-fold cross validation, train/test splitting may cause overfit.

Author response:  We appreciate the reviewer’s concern in this regard. The pediatric CXR dataset is made available by the authors of [4] with a defined patient-specific train/test split. We preferred to use this patient-specific split to evaluate for performance improvement.

We regret the lack of clarity because we missed including the statement that we randomly allocated 10% of the training data to validate the DL models. Callbacks are used to monitor model performance with the validation data and store the best model weights for further analysis with the hold-out test data. We have included these details in the revised manuscript, as shown below:

Table 1 shows the distribution of data extracted from the datasets identified above and used for the different stages of learning performed in this study. The numerator and denominator show the number of train and test data used in models’ training and evaluations. We randomly allocated 10% of the training data to validate the DL models.

Callbacks are used to monitor model performance with the validation data and store the best model weights for further analysis with the hold-out test data.

We agree that we did not perform cross-validation studies; however, we worked in several ways to see that the models did not overfit to the training data. This includes comparing the model results in a blinded fashion to the GT annotations provided by an expert radiologist. It is evident from the salient ROI localization shown in Figure 6 that the best-performing model learns the implicit rules to generalize well and conform to the experts’ knowledge about the problem.

Q8. Minor points: “Callbacks are used to monitor model performance and store the best model weights for further analysis.” This may cause overfit if authors monitored performance in test data of COIVT-19.  

Author response:  Agreed. We regret the lack of clarity in this regard. We missed including the statement that we randomly allocated 10% of the training data to validate the DL models. Callbacks are used to monitor model performance with the validation data and store the best model weights for further analysis with the hold-out test data. We have included these details, as shown below:

Table 1 shows the distribution of data extracted from the datasets identified above and used for the different stages of learning performed in this study. The numerator and denominator show the number of train and test data used in models’ training and evaluations. We randomly allocated 10% of the training data to validate the DL models.

Callbacks are used to monitor model performance with the validation data and store the best model weights for further analysis with the hold-out test data.

Q9. “On the other hand, GAN-based DL models that are used  for  synthetic  data  generation  are  computationally  complex  and  the  jury  is  still  out  on  the  anatomical and pathological validity of synthesized images.” Recently-proposed data augmentation methods should be referred to. Such as, mixup, random erasing, ricap, and so on. In addition, papers using these recently-proposed data augmentation methods and medical images should be cited.

Author response:  We render our sincere thanks to the reviewer for his valuable comments. In this regard, we have included the following reference to the revised manuscript:

[9]. Takahashi, R.; Matsubara, T.; Uehara, K. Data Augmentation using Random Image Cropping and Patching for Deep CNNs. arXiv Preprint arXiv: 1811.09030, 2019.

This is a great study where the authors proposed a novel augmentation method based on random image cropping and patching (RICAP). The proposed approach randomly crops a set of four images to create a patch, thereby mixing four class labels to result in label smoothing and further augment the training data to improve prediction performance. The authors obtain superior performance with CIFAR-100 and ImageNet classification tasks. We have discussed this reference, as shown below:

Other methods such as random image cropping and patching (RICAP) [9] are proposed for natural images to augment the training data to achieve superior performance on CIFAR-100 and ImageNet classification tasks.

Q10. “These  include  traditional  augmentation  techniques  like  flipping, rotations,  color  jittering,  random  cropping,  and  elastic  distortions  and generative  adversarial networks (GAN) based synthetic data generation [8].”  Ref. 8 is not related with medical images. Please cite papers where GAN was used for data augmentation of medical images.

Author response:  Agreed. We have removed that citation and included another that used state of the art Progressive GANs for augmenting medical images. The included reference is shown below:

Ganesan, P.; Rajaraman, S.; Long, R.; Ghoraani, B.; Antani, S. Assessment of Data Augmentation Strategies Toward Performance Improvement of Abnormality Classification in Chest Radiographs. Proc. 41st Annual International Conference of the IEEE Engineering in Medicine and Biology Society (EMBC), 2019, 841-844.

Reviewer 2 Report

The manuscript is well written and comprehensible. The study design is clear and results are clearly described.
The discussion and the conclusions follow the results.
The quality of tables and figures is fine.
References are updated.

Author Response

Q1. The manuscript is well written and comprehensible. The study design is clear and results are clearly described. The discussion and the conclusions follow the results. The quality of tables and figures is fine. References are updated.

Author response:  We render our sincere thanks to the reviewer for his valuable time and comments toward our study. To our best knowledge, we have addressed the concerns of all the reviewers so that the revised manuscript is considered acceptable for publication.

Reviewer 3 Report

This is an interesting and topical paper that deserves publication. The paper is clear and concise. I only have a few comments that the authors could address in a minor revision:

  1. The analysis provided by the authors concludes that the use of the weakly labelled datasets does not improve Covid19 detection dramatically. Some improvement is observed when the CheXpert data is used, but the accuracy is still low. Can the authors expand in their Discussion/Conclusion section on how they think this method can lead to clinically useful results?
  2. Further to the previous comment, can the authors add some comment on how this method compares to their recently proposed approache in Ref. [6]?
  3. Finally, can the authors include a discussion in the Introduction about how successful DL methods have been in detecting Covid19 for CT rather than CXR scans? Ref. [3] is certainly relevant, and so are many other papers that have emerged since the outbreak, e.g. https://pubs.rsna.org/doi/10.1148/radiol.2020200905    

Author Response

Q1. This is an interesting and topical paper that deserves publication. The paper is clear and concise. I only have a few comments that the authors could address in a minor revision: The analysis provided by the authors concludes that the use of the weakly labelled datasets does not improve Covid19 detection dramatically. Some improvement is observed when the CheXpert data is used, but the accuracy is still low. Can the authors expand in their Discussion/Conclusion section on how they think this method can lead to clinically useful results?

Author response:  We sincerely thank the reviewer for his valuable time, effort, and constructive comments toward our study. To our best knowledge, we have addressed the reviewer’s concerns to make the revised manuscript acceptable for publication. In response to this query, we have included the following discussion on how the proposed approach could be useful toward clinical assessment:

Weakly labeled data augmentation helped improve performance with the hold-out baseline test data because the CXRs with pneumonia-related opacities in CheXpert collection has a similar distribution to bacterial and non-COVID-19 viral pneumonia. This similarity helped expand the training feature space by introducing a controlled class-specific feature variance that improves performance with the baseline test data. However, with COVID-19 CXRs, weakly-labeled data augmentation didn’t deliver superior performance on its own primarily due to small data set size – which is the base reason for relying on other datasets – and distinct opacity patterns compared to non-COVID-19 viral and bacterial pneumonia. As mentioned before, the resulting classifier can recognize COVID-19, but unable to distinguish between other opacities and normal CXRs. In clinical use, it could quickly help separate patients with COVID-19 opacities (true positives) and refer the rest for further clinical assessment.

Q2. Further to the previous comment, can the authors add some comment on how this method compares to their recently proposed approach in Ref. [6]?

Author response:  Agreed. We have included the following discussion to the revised manuscript:

This discriminative training data augmentation strategy recognizes the biological similarity in viral pneumonia and radiological manifestation due to COVID-19 caused respiratory disease and dissimilarity to bacterial and non-COVID-19 viral pneumonia-related opacities. In this work, we show that it is possible to obtain a classifier that uses a collection of weakly-labeled image datasets that are related to, but clinically different from, a subclass (i.e., COVID-19) and identify it with high precision. Rejects from the classifier developed in this study are not necessarily normal and should be subjected to separate clinical assessment. Conversely, our prior work proposes an iteratively pruned deep learning ensemble strategy [6] that separates normal CXRs from those with COVID-19 and other pneumonia-like opacities.

Q3. Finally, can the authors include a discussion in the Introduction about how successful DL methods have been in detecting Covid19 for CT rather than CXR scans? Ref. [3] is certainly relevant, and so are many other papers that have emerged since the outbreak, e.g. https://pubs.rsna.org/doi/10.1148/radiol.2020200905   

Author response:  We sincerely thank the reviewer for his constructive comments in this regard. We have included the reference suggested by the reviewer at [5] and discussed it in the Introduction as follows:

A study of literature shows that automated computer-aided diagnostic (CADx) tools built with data-driven deep learning (DL) algorithms using convolutional neural networks (CNN) have shown promise in detecting, classifying, and quantifying COVID-19-related disease patterns using CXRs and CT scans [2, 3, 5, 6] and can serve as a triage under resource-constrained settings thereby facilitating swift referrals that need urgent patient care.

Round 2

Reviewer 1 Report

Based on the revision, I assume the following five points for my review comments;

  1. Normal cases were ignored in this study.
  2. For using weakly-labeled data augmentation, VP and CP were handled with the same class.
  3. CP images was not used as the training data of deep learning, except for Table 7.
  4. In Table 6, the deep learning model classified test data including CP images into BP or VP. If the model could classify CP images into VP, the model’s classification was considered as correct.
  5. The authors used supervised learning for the deep learning model.

Major points

Because of my point (I), clinical usefulness of this study is very limited. Generally, CXR is used for screening or disease monitoring. To differentiate types of pneumonia, chest CT is generally used instead of CXR. Maybe, this manuscript is suitable for other journals.

If my five points are correct, please emphasize my five points in Abstract, Introduction, and Conclusion more strongly.

"2.4. Weakly‐labeled Data Augmentation" Even after the revision, usage of UP images of RSNA, CheXpert, and NIH was not clear. Under supervised learning, please describe the way to utilize UP images as weakly-labeled data augmentation more clearly.

“Conversely, our prior work proposes an iteratively pruned deep learning ensemble strategy [6] that separates normal CXRs from those with COVID-19 and other pneumonia-like opacities.” As shown in my previous comment, I cannot accept ref. 6 because ref. 6 is preprint. Please cite ref. 6 after formally published.   

I speculate that Table 1 is wrong. If Table 1 was correct, images of RSNA, CheXpert, and NIH were used as test data. According to Tables 5 and 6, images of RSNA, CheXpert, and NIH must be used as training data. In Table 1, please describe the total number of CXR images used in this study.

Please clarify the way to select CXR images of NIH and CheXpert used in this study. These datasets have more than 100000 CXR images.

For each (combined) dataset of Tables 3, 5, and 6, please describe the numbers of UP, BP, VP, and CP images used as training data and test data.

If my point (II) is correct, Figure 2 is wrong. C1 and C2 are BP and VP (VP including CP).

Minor points

Words in Figure 4 and 6 are too small.

In Table 7 and Figure 6, I speculate that weakly-labeled data augmentation was not used. Therefore, please delete them or move them to Appendix.

Author Response

Q1: Based on the revision, I assume the following five points for my review comments;

1(a): Normal cases were ignored in this study.

Author response:  Agreed.

1(b): For using weakly-labeled data augmentation, VP and CP were handled with the same class. CP images was not used as the training data of deep learning, except for Table 7.

Author response:  Agreed.

1(c): In Table 6, the deep learning model classified test data including CP images into BP or VP. If the model could classify CP images into VP, the model’s classification was considered as correct.

Author response:  Agreed.

1(d): The authors used supervised learning for the deep learning model.

Author response:  Agreed.

1(e): Because of my point (1(a)), clinical usefulness of this study is very limited. Generally, CXR is used for screening or disease monitoring. To differentiate types of pneumonia, chest CT is generally used instead of CXR. Maybe, this manuscript is suitable for other journals.

Author response:  We appreciate the reviewer’s concern in this regard. As we elaborated in our previous response, the methods proposed in this manuscript are not intended to be viewed as a system for screening patients. Rather, as the title suggests, it is a study in the use of weakly labeled sets to devise a highly selective classifier (identifying capability) of a particular class or subclass. The urgency and lack of widely available datasets for the COVID-19 pandemic provided this unique opportunity to test our hypothesis for a meaningful cause. The hypothesis is that if we have a large collection of weakly labeled image datasets that are related to but clinically different from a subclass, it is possible to develop a classifier that identifies these (i.e., subclass) images with high precision. Rejects from this classifier are not necessarily normal and should be subjected to separate clinical assessment.

While chest X-rays and computed tomography (CT) scans have been used to screen for COVID-19 and/or other bacterial and viral pneumonia infections and evaluate disease progression in hospital admitted cases [1] [2], indeed they are not currently recommended as primary diagnostic tools.  While chest CT offers greater sensitivity to pulmonary disease detection, there are several challenges to its use. These include the non-portability, the requirement to sanitize the room and equipment between patients followed by a delay of at least an hour [1], the risk of exposing the hospital staff and other patients, and persons under investigation (PUI) to the virus. The American College of Radiology has also supported the above observation (https://www.acr.org/Advocacy-and-Economics/ACR-Position-Statements/Recommendations-for-Chest-Radiography-and-CT-for-Suspected-COVID19-Infection) and recommends the use of CXRs as an alternative imaging technique when deemed necessary. Although not as sensitive, portable CXRs are used [1] since the PUIs can be imaged in more isolated rooms using mobile X-ray equipment with plastic-covered digital X-ray plates, and limit personnel exposure. Thus, sanitation needs are much less complex to obtain than with CT. Recent studies have shown that (https://www.auntminnie.com/index.aspx?sec=sup&sub=aic&pag=dis&ItemID=128976) a commercial artificial intelligence (AI) software application is using chest radiographs in triaging suspected COVID-19 cases.

 Q2: If my five points are correct, please emphasize my five points in Abstract, Introduction, and Conclusion more strongly.

Author response:  Agreed and modified. The changes are highlighted in yellow in the revised manuscript.

Q3: "2.4. Weakly‐labeled Data Augmentation" Even after the revision, usage of UP images of RSNA, CheXpert, and NIH was not clear. Under supervised learning, please describe the way to utilize UP images as weakly-labeled data augmentation more clearly.

Author response:  We regret the lack of clarity in this regard. The manuscript is revised to include the following: Table 1, Table 2, and Table 3 in the revised manuscript show the distribution of data used toward baseline training and evaluation, weak-label augmentation, and COVID-19 classification, respectively. We limit our study only to pneumonia cases of four varieties: unknown type, bacterial, viral, or COVID-19. Though the CheXpert dataset contains 223,648 CXRs, the metadata only identifies 4683 CXRs as containing pneumonia-related opacities. The same holds for the RSNA and NIH CXR collections which contain 6012 and 307 CXRs only that are labeled as “pneumonia” without further qualification. We treat these also as belonging to the “unknown type”. The resulting “unknown type” pneumonia collection is thus (6012 + 4683 + 307 = 11,002) CXR images.

Table 1. Baseline dataset characteristics. Numerator and denominator denote the number of train and test data respectively (BP= Bacterial (proven) pneumonia, VP= non-COVID-19 viral (proven) pneumonia).

Dataset

BP

VP

Pediatric

2538/242

1345/148

Table 2. Characteristics of datasets used for weak-label classification. (UP=Pneumonia of unknown type).

Dataset

UP

RSNA

6012

CheXpert

4683

NIH

307

Table 3. Distribution of COVID-19 CXR data. (CP = COVID-19 pneumonia).

Dataset

CP

Twitter-COVID-19

135

Montreal-COVID-19

179

Fig. 2 in the revised manuscript illustrates the modified graphical abstract of the proposed study. Broadly, our workflow consists of the following steps: First, we preprocess the images to make them suitable for use in DL. Then, we evaluate the performance of a custom CNN and a selection of pre-trained CNN models for categorizing the pediatric CXR collection, referred to as baseline, as showing bacterial or viral pneumonia, as shown in Fig. 2a. The trained model is further used to categorize the publicly available COVID-19 CXR collections as showing viral pneumonia. Next, we use the top-performing model from (a) to weakly label CXRs in the publicly available RSNA, CheXpert, and NIH collections manifesting pneumonia-related opacities of the bacterial or viral type as shown in Fig. 2b. Next, the baseline training data is augmented with these weakly labeled CXRs toward improving detection performance, as shown in Fig. 2c. This discriminative training data augmentation strategy recognizes biological similarity in viral and COVID-19 pneumonia, i.e. both are viral; however, also note distinct radiological manifestations between each other as well as with non-viral pneumonia-related opacities. As suggested, we have added clarity to the weakly-labeled data augmentation Section in the revised manuscript with highlighted changes.

Q4:  “Conversely, our prior work proposes an iteratively pruned deep learning ensemble strategy [6] that separates normal CXRs from those with COVID-19 and other pneumonia-like opacities.” As shown in my previous comment, I cannot accept ref. 6 because ref. 6 is preprint. Please cite ref. 6 after formally published.  

Author response:  Removed as requested.

Q5: I speculate that Table 1 is wrong. If Table 1 was correct, images of RSNA, CheXpert, and NIH were used as test data. According to Tables 5 and 6, images of RSNA, CheXpert, and NIH must be used as training data. In Table 1, please describe the total number of CXR images used in this study.

Author response:  We regret the lack of clarity. We have made changes to the Tables as shown in our response to Q3.

Q6: Please clarify the way to select CXR images of NIH and CheXpert used in this study. These datasets have more than 100000 CXR images.

Author response:  Thanks. We refer the reviewer to our response to Q3. We limit our study only to pneumonia cases of four varieties: unknown type, bacterial, viral, or COVID-19. Though the CheXpert dataset contains 223,648 CXRs, the metadata only identifies 4683 CXRs as containing pneumonia-related opacities. The same holds for the RSNA and NIH CXR collections which contain 6012 and 307 CXRs only that are labeled as “pneumonia” without further qualification. We treat these also as belonging to the “unknown type”. The resulting “unknown type” pneumonia collection is thus (6012 + 4683 + 307 = 11,002) CXR images.

Q7: For each (combined) dataset of Tables 3, 5, and 6, please describe the numbers of UP, BP, VP, and CP images used as training data and test data.

Author response:  Agreed. Table 5 in the revised manuscript uses data from Table 1. Table 6 uses data from Table 3. Table 7 shows the number of samples across the bacterial and viral pneumonia categories after augmenting the baseline pediatric CXR training data with weakly-labeled images from the respective CXR collections.

Table 7. Number of samples in weakly-labeled augmented training data.

Dataset

BP

VP

Baseline + NIH

2720

1470

Baseline + CheXpert

4683

3883

Baseline + RSNA

6577

3318

Baseline + NIH + CheXpert

4865

4008

Baseline + NIH + RSNA

6759

3443

Baseline + CheXpert + RSNA

8722

5856

Baseline + NIH + CheXpert + RSNA

8904

5981

Table 8 in the revised manuscript uses data from Table 7 for training and the test data from Table 1.  

Table 9 uses data from Table 7 for training, and Table 3 for testing.

Q8: If my point (II) is correct, Figure 2 is wrong. C1 and C2 are BP and VP (VP including CP).

Author response:  Modified as suggested. We refer the reviewer to our response to Q3 concerning the changes made to Figure 2.

Q9: Minor points: Words in Figure 4 and 6 are too small. Modify them. Put them in separate rows.

Author response:  Modified as suggested.

Q10: In Table 7 and Figure 6, I speculate that weakly-labeled data augmentation was not used. Therefore, please delete them or move them to Appendix.

Author response:  We appreciate the reviewer’s concern in this regard. The results discussed in Table 10, and Table 11 in the revised manuscript are the very interesting observations from this study. We have moved these analyses to a separate Discussion section (Section 4), as suggested.

For clarity – weakly-labeled augmentation is very valuable when there are insufficiently large training datasets. This is observed by the best-performing training data combination in Table 8 (i.e., Baseline + CheXpert). Typically this would have been the end-point of our study. However, from our results in Table 6, we saw that a viral pneumonia classifier was unable to detect COVID-19. Also, the results – while better – did not gain significantly from weakly-labeled augmentation as shown in Table 9. These results implied that COVID-19 was not captured with the variety of viral pneumonia in other datasets.

To test this idea, we added COVID-19 data to the best performing weakly-labeled augmented training data (Baseline + CheXpert) (Table 10); and, also directly to the baseline (Table 11). These findings add value to the study on COVID-19, as much as weakly-labeled data augmentation adds value to improving results when there is insufficient data, in general. These points are elaborated in Discussion (Section 4).

We hope that the reviewer agrees to our responses in this regard. We sincerely thank the reviewer for the valuable time, effort, and insightful comments that helped to revised this manuscript and make it suitable for possible publication. 

Round 3

Reviewer 1 Report

Major points

Authors’ replay “As we elaborated in our previous response, the methods proposed in this manuscript are not intended to be viewed as a system for screening patients. Rather, as the title suggests, it is a study in the use of weakly labeled sets to devise a highly selective classifier (identifying capability) of a particular class or subclass.” I agree with this opinion. Because of this authors’ replay, I think that this paper focused on the technical point, not on the clinical one. Therefore, this paper may be suitable for other journals.

“2.4. Weakly-labeled Data Augmentation” Based on the Figure 2, I assume the following two points. (I) UP images were classified into BP or VP by the DL model trained on pediatric CXR. (II) The classification results of (I) were used as ground truth labels of weakly-labeled augmented training data for UP images. If these two points are correct, please clarify these two points in 2.4.  

Minor points

In Figures 4 and 7, font is too small after the revision.

Author Response

Q1: Authors’ replay “As we elaborated in our previous response, the methods proposed in this manuscript are not intended to be viewed as a system for screening patients. Rather, as the title suggests, it is a study in the use of weakly labeled sets to devise a highly selective classifier (identifying capability) of a particular class or subclass.” I agree with this opinion. Because of this authors’ replay, I think that this paper focused on the technical point, not on the clinical one. Therefore, this paper may be suitable for other journals.

Author response:  We are happy to note that the reviewer recognized that this work was technical in nature and not clinical. We do not claim that this work should be directly machined for field use. That said, we believe that our work adds to the science in understanding the effect of training data variety in designing classifiers which, in turn, could impact diagnostic performance. Our work also helps shape future efforts in recognizing COVID-19 pneumonias when more data might become available, and one has the luxury of designing classifiers to detect these as abnormal chest X-rays and separating them from normal and other pneumonia categories.

Q2: “2.4. Weakly-labeled Data Augmentation” Based on the Figure 2, I assume the following two points. (I) UP images were classified into BP or VP by the DL model trained on pediatric CXR. (II) The classification results of (I) were used as ground truth labels of weakly-labeled augmented training data for UP images. If these two points are correct, please clarify these two points in 2.4. 

Author response:  Agreed. We modified the Tables 1, 2, and 3 to remove abbreviations and convey clarity. Section 2.4 is modified to include reviewer’s suggestions.

Q3: In Figures 4 and 7, font is too small after the revision.

Author response:  Modified as suggested.